# Pilot Study Showing Feasibility of Phosphoproteomic Profiling of Pathway-Level Molecular Alterations in Barrett’s Esophagus

**DOI:** 10.3390/genes13071215

**Published:** 2022-07-07

**Authors:** Jarrod Moore, Ryan Hekman, Benjamin C. Blum, Matthew Lawton, Sylvain Lehoux, Matthew Stachler, Douglas Pleskow, Mandeep S. Sawhney, Richard D. Cummings, Andrew Emili, Alia Qureshi

**Affiliations:** 1Center for Network Systems Biology, Boston University School of Medicine, Boston, MA 02118, USA; jmoore5@bu.edu (J.M.); rhekman@bu.edu (R.H.); bblum@bu.edu (B.C.B.); mllawton@bu.edu (M.L.); aemili@bu.edu (A.E.); 2Harvard Medical School Center for Glycoscience, Harvard Medical School, Boston, MA 02115, USA; sylv.lehoux@gmail.com (S.L.); dpleskow@bidmc.harvard.edu (D.P.); rcummin1@bidmc.harvard.edu (R.D.C.); 3Helen Diller Family Comprehensive Cancer Center, University of California San Francisco, San Francisco, CA 94158, USA; matthew.stachler@ucsf.edu; 4Department of Surgery, Harvard Medical School, Boston, MA 02115, USA; msawhney@bidmc.harvard.edu; 5Division of General & GI Surgery, Oregon Health & Science University, Machall 3186, Portland, OR 97239, USA

**Keywords:** pre-cancerous lesion, biopsy, systems biology, mass spectrometry, disease signature

## Abstract

(1) Background: Barrett’s esophagus is a major risk factor for esophageal adenocarcinoma. In this pilot study, we employed precision mass spectrometry to map global (phospho)protein perturbations in Barrett’s esophagus lesions and adjacent normal tissue to glean insights into disease progression. (2) Methods: Biopsies were collected from two small but independent cohorts. Comparative analyses were performed between Barrett’s esophagus samples and adjacent matched (normal) tissues from patients with known pathology, while specimens from healthy patients served as additional controls. (3) Results: We identified and quantified 6810 proteins and 6395 phosphosites in the discovery cohort, revealing hundreds of statistically significant differences in protein abundances and phosphorylation states. We identified a robust proteomic signature that accurately classified the disease status of samples from the independent patient cohorts. Pathway-level analysis of the phosphoproteomic profiles revealed the dysregulation of specific cellular processes, including DNA repair, in Barrett’s esophagus relative to paired controls. Comparative analysis with previously published transcriptomic profiles provided independent evidence in support of these preliminary findings. (4) Conclusions: This pilot study establishes the feasibility of using unbiased quantitative phosphoproteomics to identify molecular perturbations associated with disease progression in Barrett’s esophagus to define potentially clinically actionable targets warranting further assessment.

## 1. Introduction

Barrett’s esophagus (BE) is a major risk factor predisposing individuals to esophageal adenocarcinoma (EAC), a highly morbid disease with poor survival despite invasive intervention [1,2,3]. BE refers to a metaplastic change in the stratified squamous epithelium lining the distal esophagus, primarily driven by chronic gastric and bile reflux, resulting in replacement with an intestinal-like columnar epithelium. Following endoscopic biopsy, a diagnosis of BE is made by expert pathologists, with the presence of goblet cells being the pathognomonic finding [4,5,6].

It has been postulated that BE is the precursor to dysplastic changes that eventually progress to EAC [2]. In one model analogous to the adenocarcinoma sequence reported for colon cancer [7], EAC progresses in a linear, stepwise sequence [4]. Patients first develop BE because of chronic acid reflux and subsequent esophagitis; over time, this lesion advances to higher grades of dysplasia until ultimately becoming overt EAC [2]. Recent applications of genomic and proteomic technologies, capable of measuring the expression of genes, proteins, and pathways differentially regulated during cancer progression, have refined this model by measuring relative alterations throughout pathogenesis [3]. For example, paired exome analysis of BE and adenocarcinoma suggests that progression to malignancy occurs via a transition driven by the amplification of specific oncogenes (including early non-dysplastic tissue) [3]; other studies have distinguished rapid, catastrophic chromosomal events as inciters of EAC [3]. Etiological evidence demonstrates that only ~0.1–1% of BE cases progress to EAC, suggesting individual heterogeneity [8]. Accordingly, the framework of pathogenesis underlying the phenotypic variation in BE patients poses significant challenges for defining consistent perturbations causal to clinical outcomes.

Mapping the maladaptive molecular responses perturbed in early-stage, non-dysplastic BE relative to normal squamous tissue could inform on the pathobiological processes preceding and driving dysplasia. Changes in gene expression, translation, and post-transcriptional regulation leading to alterations in protein abundance are thought to drive key aspects of pathobiology, while phosphorylation-based signal transduction pathways are likely to control key cellular processes linked to dysplasia. Nevertheless, few unbiased phosphoproteomic profiling studies on clinical samples have been reported to date [9].

In this pilot study, we systematically applied deep quantitative proteomic and phosphoproteomic profiling to a select set of case and control clinical samples, in conjunction with rigorous statistical analysis, to define a preliminary set of differential proteins, phospho-sites, and pathways between non-dysplastic BE and adjacent squamous tissue in multiple affected patient cohorts. We identified candidate molecular markers and pathways distinguishing BE from normal tissue, whose association suggests potential roles in pre-malignant disease progression. These preliminary findings were supported by unsupervised classification of several independent sets of patient specimens. Moreover, we completed gene set enrichment analyses on phosphoproteome, proteome, and transcriptome data to glean insights into pathway-level perturbations in BE.

## 2. Materials and Methods

Sample Acquisition and Identification: Forty paired biopsy specimens representing BE lesions, adjacent (normal, squamous epithelium extracted within 2 mm from respective BE lesions), and non-BE (squamous epithelium from patients without BE pathology) tissues were obtained from ten patients (five with and five without confirmed BE) receiving care at Beth Israel Deaconess Medical Center in Boston, MA (see Appendix A). Patients were informed of the study aims and enrolled in an IRB-approved protocol (2017P000203), approved by the Beth Israel Deaconess Medical Center Committee on Clinical Investigations. Patients were enrolled into their respective groups based on previous BE diagnosis (i.e., non-BE patients had no previous diagnosis and were undergoing endoscopy for symptoms, while BE patients had a BE diagnosis). Paired biopsy samples were processed as two separate cohorts (10 specimens per cohort): namely, an initial discovery cohort (3 BE patients and 2 controls) and an independent validation cohort (2 BE patients and 3 controls), both of which were subjected to quantitative proteomic/phosphoproteomic analysis by quantitative LC/MS (Figure 1, created with BioRender.com. Accessed on 8 May 2022).

Tissue Homogenization and Trypsin Digestion: Paired samples (~1.5 × 2 mm) were thawed in 150 µL of lysis buffer containing 8 mol/L urea, 5 mmol/L DTT, 50 mmol/L NH_4_HCO_3_, and Protease (cOmplete) and phosphatase inhibitors (PhosStop; Roche, Basel, Switzerland), mechanically homogenized (MM400, Retsch, Haan, Germany), and sonicated (Branson) on ice. Soluble protein (200 µg; Bradford) was alkylated by the addition of iodoacetamide (15 mmol/L). After incubation at room temperature for 1 h in the dark, samples were quenched with excess DTT and diluted with 1400 µL of NH_4_HCO_3_ (bringing urea to 1.5 mol/L) followed by digestion for ~17 h at 37 °C with 5 µg of sequencing-grade Trypsin (ThermoFisher, Waltham, MA, USA). After adding trifluoroacetic acid to 0.1% *v*/*v*, peptide digests were desalted, resuspended in 100 mmol/L TEAB, and quantified by Quantitative Colorimetric Peptide Assay (Pierce).

Stable-Isotope Labeling and Offline Reverse-Phase High-Performance Liquid Chromatography: For each cohort, 100 µg of each biopsy peptide digest (adjusted to 100 µL with 100 mmol/L TEAB) was mixed with a unique amine-reactive isotope-coded isobaric tandem mass tag (TMT-11plex) reagent (ThermoFisher) prior to sample multiplexing and precise quantification by LC/MS. After pooling, 1 mg of total labeled peptide was desalted, dried, and suspended in 300 µL of buffer containing 0.1% ammonium hydroxide and 2% ACN. For the discovery cohort, the pooled sample mixture was pre-fractionated by high pH reverse-phase HPLC (XBridge Peptide BEH C18, 130 Å, 3.5 μm, 4.6 mm × 250 mm, Waters, Milford, MA, USA) using an Agilent 1100 pump. Peptides were eluted using a gradient of mobile phase A (2% ACN and 0.1% NH_4_OH) to B (98% ACN and 0.1% NH_4_OH) over 48 min and collected as 12 pooled fractions. For phosphoproteomics, the bulk (95%) of each sample was subjected to phosphopeptide enrichment using TiO2 metal-chelate resin (Titansphere Phos-TiO, 10 µm, GL Sciences, Shinjuku-ku, Tokyo, Japan) [10], while the remaining (5%) portions were analyzed directly by nanoflow LC/MS as bulk proteome measurements (24 injections for the discovery cohort and 2 for the validation cohort).

Mass Spectrometry and Peptide Identification: Isotope-labeled peptides were reconstituted in solvent (0.1% formic acid and 2% ACN) prior to LC/MS analysis on a ThermoScientific Q-Exactive HFX mass spectrometer connected to an EASY-nLC-1200 ultra-high-pressure nanoflow chromatography system. After loading onto a reverse-phase trap (75 μm i.d. × 2 cm, Acclaim 3 μm 100 Å PepMap100-C18 resin, ThermoScientific, Waltham, MA, USA), peptides were gradient-separated on an EASY-Spray C18 nanocolumn (75 μm i.d. × 50 cm, 2 μm, 100 Å; ES803A, ThermoScientific) using 2–35% B (0.1% formic acid and 80% ACN) over 120 min and electro-sprayed at ~250 nL/min into the HFX instrument in positive ion mode (capillary temperature 275 °C; 2100 V potential). Data-dependent spectra were acquired automatically via high-resolution (60,000) precursor ion scans (350–1500 *m/z* range) to select the 12 most intense peptide ions for MS/MS fragmentation by high-energy dissociation using a normalized collision energy of 33 at 45,000 resolution.

The resulting RAW files were searched by MaxQuant (software version number 1.6.7.0, Jürgen Cox, Max Planck Institute of Biochemistry, Planegg, Germany) using default settings against the human proteome (SwissProt Taxonomy ID: 9606, downloaded on the 10 April 2019), allowing for two missed cleavage sites and variable modifications (Ser/Thr/Tyr phosphorylation, N-terminal acetylation, and Met oxidation) and carbamidomethylation of cysteine and TMT labels as a fixed modification. Peptide- and protein-level matches were filtered to high confidence (1% FDR), with a minimum phosphosite localization probability of 0.7. TMT quantification involved label lot value correction.

Statistical Analysis and Pathway Enrichment: Bioinformatic analysis was performed using R (language and environment for Statistical Computing; http://www.R-project.org. Accessed on the 25 July 2020). Peptide feature intensities were log-transformed and quantile-normalized. The LIMMA [11] R package was used to perform differential analysis (moderated Student *t*-tests) and to generate ranked lists for subsequent enrichment analysis using the Benjamini–Hochberg FDR correction. For the BE signature, empirical FDR (<0.1) was based on 100-fold randomization. Statistical enrichment analysis was performed using the fgsea R package [12,13] with 10,000 permutations to define empirical FDR cutoffs (see Appendix A). Published transcriptomic datasets (raw expression values and corresponding gene names) were obtained from GEO using the GEOquery R package and are referred to as Nancarrow [14] (22 BE/9 normal), Hyland [15] (40 BE/40 matched normal), and Stairs [16] (7 BE/7 matched adjacent tissue). Volcano plots were created using the EnhancedVolcano R package [17].

Sex-Inclusive Biomedical and Clinical Research: The ratio of females to males in this study is 3:2 (6 females and 4 males).

## 3. Results

### 3.1. Comparative Proteomic Profiling Reveals a Differential Disease Signature Reproducibly Associated with BE

The proteomic workflow is illustrated in Figure 1. Briefly, we performed quantitative LC/MS-based proteomic and phosphoproteomic profiling on 10 paired biopsy samples (5 non-dysplastic biopsies and 5 paired adjacent, normal squamous epithelium biopsies) from 5 BE patients and 10 “control” biopsies from 5 non-BE endoscopy patients (Figure 1a). These samples were split and analyzed as two groups: an initial discovery set (cohort 1) and a second validation set (cohort 2). To ensure quantitative accuracy, the two sets of samples were subjected to stable isotope (isobaric tandem mass tag) labeling prior to precision LC/MS. To augment proteome coverage, the discovery cohort specimens were also first subjected to extensive pre-fractionation before the LC/MS analysis (Figure 1b).

In total, we identified and quantified 6810 distinct proteins in the discovery cohort and 2993 in the validation cohort (Appendix A). Comparative analysis revealed hundreds of statistically significant differences (*p* < 0.05, moderated *t*-test; FDR < 0.1 based on randomization) in protein relative abundance between both sets of paired BE and adjacent biopsy samples (Figure 2a). Of note, we were able to capture multiple proteins associated with BE from prior comparative studies, including mucin and mucin-related proteins (MUC5AC and TFF1) [18].

To define putative disease-specific expression patterns, we applied moderated *t*-tests to define the top differentially expressed proteins and phosphosites between the two pathobiological classes from our discovery cohort. We selected the top 1% of features that also showed at least +/− 0.75 log2-fold change, a rigorous threshold given documented TMT ratio compression [19,20]. From the proteome set, we identified 59 differential proteins, 13 with elevated expression in BE and 46 with reduced expression (Figure 2a and Appendix A). These top differential features represent putative molecular markers for BE and highlight important components of the cell proliferation, DNA repair response, and cell cycle machinery. For example, BE samples had increased expression of HORMA domain-containing protein 1 (HORMAD1), an essential component of cell cycle regulation [21,22]. Interestingly, mRNA data suggest that overexpression of this gene is highly specific for gastric cancer, with over 40% of survey samples showing a 2-fold increase in expression [23].

### 3.2. Validation of BE Expression Signature in a Second Patient Cohort

To ensure rigor, given the small discovery cohort, we sought to replicate our putative BE signature by independently confirming the differential abundance of these same putative markers in association with disease status in the separate validation biopsies and by comparison to non-BE (healthy) patient samples. Volcano plots displaying the relative protein expression of just the subset of candidate signature proteins reproducibly detected in the second cohort showed substantive overall agreement in disease-specific expression (BE vs. adjacent) across both BE patient groups (Figure 2a). For example, FBLN5 and MFAP4 were reproducibly observed to be significantly differentially expressed in the independent validation set of BE lesions, while CRNN and KRT6B were once again preferentially detected in the adjacent control tissue specimens (Appendix A).

As an additional test to assess whether the inferred BE signature (derived from the discovery cohort) alone could correctly classify disease status (lesions versus adjacent normal or healthy tissue), we used hierarchical agglomerative clustering to group the validation patient samples according to their relative expression of the BE markers that were detected in the validation cohort. This classifier correctly classified the patient specimens based on disease status, as evidenced by sample hierarchical clustering into respective disease status groups (Figure 2b). The clear separation of BE, adjacent, and healthy tissues based solely on their proteomic profile similarity indicates a high overall classification accuracy and the pertinence of these features.

Finally, as a further assessment of reliability, we projected our proteomics-derived BE signature against an independent transcriptomic dataset encompassing the corresponding gene expression profiles recorded for 7 BE and matched normal tissues (14 samples in total) from the Stairs et al. publication [16] to verify the specificity of our classifier. While the correspondence of protein and cognate mRNA patterns can be uneven, numerous comparative studies have shown a positive overall correlation between corresponding marker transcript and protein levels [24,25]. Notably, hierarchical clustering of the Stairs et al. clinical samples based only on the corresponding messenger RNA levels that match our predicted signature proteins was able to faithfully classify this independent set of BE and adjacent normal (squamous epithelium) samples into distinct clusters reflective of disease status (Figure 2b).

Based on these extensive confirmatory results, we concluded that this preliminary set of putative BE markers was enriched for proteins that are predictive molecular indicators of pathology.

### 3.3. Comparative Proteomic Analysis Reveals Pathway-Level Alterations in BE

To assess the broader functional significance of the global proteomic differences that we measured in the discovery cohort, we performed a statistical overrepresentation analysis (GSEA) to assess the molecular pathways altered in the diseased state relative to the control tissue. Our analysis revealed an extensive set (186) of significantly enriched (FDR < 0.05) pathways demarcating intra- and extracellular processes. These could be classified into four broad annotation categories (Figure 3a): (1) extracellular matrix remodeling and epithelial cell differentiation, (2) immune activity, (3) genomic integrity and cell stress, and (4) protein expression and cell metabolism (Appendix A).

For example, BE lesions were enriched for the cellular response to unfolded protein (FDR = 0.016), an endoplasmic reticulum stress pathway that is activated in human esophagus cells treated with bile acids (Figure 3a,b) [26]. Components of this pathway include mitochondrial heat shock proteins and stress-associated endoplasmic reticulum factors involved in responding to unfolded proteins. Conversely, the adjacent normal control samples were enriched for pathways involved in translation initiation (FDR = 0.013) and elongation, suggesting a defect in protein synthesis in the lesions.

The BE samples were also enriched for elevated components linked to the epithelial-to-mesenchymal transition (EMT, FDR = 0.013), suggesting an association of this non-dysplastic phenotype with tissue invasion. Key components detected included actin aortic- smooth muscle and fibulin-5, as well as Vimentin and S100-A4 (Appendix A) [27,28,29]. Supplementing this, we detected decreased enrichment of epithelial differentiation factors (FDR = 0.013) in the BE lesions, along with increased levels of extracellular matrix modification factors (FDR = 0.013).

Concomitantly, we observed increased immune-related pathways in the BE samples, including enrichment for chemokine cascades, such as inflammation mediated by chemokine/cytokine signaling (FDR = 0.049) and leukocyte activation (FDR = 0.018). Mechanisms related to innate immunity have previously been implicated in BE, with studies reporting the overproduction of interferon-γ and other inflammatory cytokines in association with more advanced disease progression and poor clinical prognosis [30,31].

### 3.4. Characterization of Differential Signaling Pathways by Phosphoproteomics

The determination of altered phospho-signaling responses provides additional functional information complementary to differential protein expression. Hence, in parallel to our proteomic survey, we used large-scale phosphopeptide affinity capture to identify and quantify 6395 phosphorylation sites on 1898 distinct phosphoproteins in the discovery cohort (Appendix A), as well as 2345 phosphosites on 743 phosphoproteins in the validation cohort (Appendix A).

As with the proteome analysis, comparative statistical analysis demonstrated hundreds of statistically significant (*p* < 0.05, moderated *t*-test) differences between the paired sample groups. As before, putative disease-specific phosphosites were defined as the top 1% of differentially abundant features between the two pathobiological classes of our discovery cohort that showed at least +/− 0.75 log2-fold change. From this stringent analysis, we identified 64 differential phosphosites, 10 showing elevated phosphorylation in BE and another 54 showing decreased levels relative to normal control tissue (Appendix A). The former included increased phosphorylation of Tumor suppressor p53-binding protein 1 (TP53BP1) on Thr1319, a double-strand repair protein that regulates genomic stability [32]. While the functional significance of these particular sites is uncharacterized, these data suggest that post-translational regulation of this key cell cycle modulator occurs preferentially in the pre-cancerous lesions. In contrast, we observed decreased phosphorylation of Cornulin (Thr268 and Ser383 on CRNN) and Zinc finger protein 185 (Thr158 and Ser307 on ZNF185), both important regulators of proliferation and cell cycle progression.

We mapped the differential phosphoproteins onto their corresponding intracellular signal transduction cascades via enrichment analysis to identify and corroborate activity between BE lesions and adjacent tissue. This revealed a set of 21 significantly altered signaling pathways (FDR < 0.05) (Figure 3a and Appendix A), many supplementing the pathway findings obtained from the proteomic analysis. These pathways mapped to diverse intra- and extracellular functional categories, with the greatest overlap with our proteomics data occurring in pathways related to epithelial development, extracellular matrix organization, and gene expression (Appendix A). Notably, the differential expression of these pathways generally shared similar normalized enrichment scores, as seen in the proteomic GSEA. For example, epithelial cell differentiation (FDR = 0.024) was enriched in adjacent normal tissue relative to BE in the phosphoproteomic GSEA as well as in the proteomic analysis (Figure 3a). Similarly, extracellular matrix organization was significantly enriched in the BE samples.

We performed motif (substrate) enrichment analysis to systematically evaluate which protein kinases are hyperactivated in BE lesions and hence potentially responsible for the differential phosphorylation-dependent signaling that we detected by phosphoproteomics. Analogous to standard pathway enrichment, this analysis projects site-specific phosphorylation data onto curated databases comprising known kinase motifs and kinase–substrate interactions, creating a list of pairwise associations (e.g., the protein kinase CSNK2A1 with its putative substrate DAXX). The significance of the associated protein kinases linked to the differential substrates identified in BE was tested by measuring the deviation from the expected background distribution, with enrichment (z-score) denoted as either positive (enriched in BE) or negative (in adjacent normal).

In total, the activities of 20 protein kinases were found to be significantly overrepresented (*p*-value < 0.05, based on motif/substrate enrichment analysis) in BE versus adjacent normal samples (Appendix A). Intuitively, these enriched kinases are predicted to be important, and possibly causal, drivers of the pre-cancerous processes associated with aberrant cell cycle progression, DNA repair, and proliferation. For example, PRKCD, which was enriched in the BE samples, is known to phosphorylate tumor suppressor p53 in response to genotoxic stress, promoting p53-dependent apoptosis [33]. Likewise, elevated phosphorylation-dependent signaling in BE by CSNK2A1, which has links to important downstream cell cycle regulator substrates such as MCM2 (Figure 4), implies potentially maladaptive (transformative) consequences in predisposing dysplastic cells towards impaired cell cycle control.

### 3.5. Pathway Analysis of Combined Cohorts via Proteomics and Phosphoproteomics

Given the observed consistency between cohorts, we combined the datasets to gain further insights into BE pathology. Phosphoproteomic and proteomic GSEA of this pooled group demonstrated similar enrichment patterns, including increased enrichment of the unfolded protein response and extracellular matrix organization in the proteome of BE samples (FDR < 0.05) (Appendix A). This analysis revealed novel pathway information. This included decreased enrichment of pathways related to the regulation of cell division, including the downregulation of components of the mitotic cell cycle, indicative of impaired cell cycle pathways associated with a loss in genomic integrity/DNA damage response and/or increased susceptibility to genetic insult. Components of this pathway included kinetochore interaction proteins (e.g., kinetochore protein Nuf2) and mitosis initiation factors, which were downregulated in our BE samples. These findings further support the genetic insult noted in the phosphoproteomic analysis of the exploratory cohort.

Additionally, we noted specific immune responses downregulated in BE samples, such as neutrophil degranulation and TNF-α signaling pathways. While we noted different enrichment in our discovery cohort analysis, our expanded analysis provided specific, differential immune pathways related to BE. Overall, this analysis showed considerable agreement with the exploratory cohort alone while providing a modest increase in pathway coverage.

### 3.6. Pathway-Level Comparison of Phosphoproteomic Profiles to Previous Transcriptomic Studies

Given the limited sample size of our pilot cohorts, we sought to ensure the broader potential relevance of these preliminary results by searching for convergent pathway-level alterations in other previously reported BE transcriptomic studies. For this, we re-analyzed extant BE transcriptome expression datasets using the same enrichment methodology that we applied to our own phospho/proteomic datasets (see Methods). In total, we analyzed three gene expression studies (named, as before, based on the first author): Stairs [16] (7 BE vs. 7 matched normal biopsies), Hyland [15] (40 BE vs. 40 normal), and Nancarrow [14] (22 BE vs. 9 normal).

Though the overall extent and directionality of the differential gene sets varied somewhat amongst the studies, the pathways enriched in the transcriptomic studies were, in general, notably similar to many of those we found significantly altered in our joint proteomic analyses (Appendix A). For instance, as with our phospho/proteomic analyses, differential analyses of transcriptome profiles revealed decreased enrichment of the mitotic spindle checkpoint (Hyland), increased enrichment of the unfolded protein response (Stairs and Hyland), and altered epithelial-to-mesenchymal transition (in all three studies). Hence, despite the widely reported inconsistency between mRNA and protein measurements and the clinical heterogeneity of BE [24,25], our proteomic analysis successfully replicated alterations in the same broad functional categories, indicating a robust set of findings pointing to fundamental and reproducible molecular perturbations in the pre-cancerous lesions.

Nonetheless, a unique advantage of our pilot study is that we have direct molecular evidence of differential protein abundance and post-translation modification, reflecting changes in protein kinase activity and their putative substrates with precise phosphosite-level resolution and quantification (Figure 3 and Figure 4). Moreover, statistical analysis of these respective datasets together further revealed a set of pathways not previously noted in any of the original studies, expanding upon the pathobiological insights of the current study. For example, our phosphoproteome data provide unique insights into signaling pathways related to these phenomena, including the related enrichment of the extracellular matrix organization pathway and decreased epithelial cell differentiation in BE samples.

### 3.7. Non-BE and Adjacent Normal Comparison Reveal Early Reactive Changes to Environmental Factors

Lastly, we compared non-BE and adjacent normal samples from both cohorts to identify early reactive changes within the non-dysplastic margins. We identified 136 differentially enriched pathways from the proteome and 29 from the phosphoproteome (FDR < 0.05). These adjacent normal samples shared many relative differences with the non-BE samples as to their respective BE counterparts (Appendix A). For example, the proteome of adjacent normal samples had decreased enrichment of DNA repair and chromatin assembly pathways with respect to the non-BE patients. Moreover, the phosphoproteome and proteome revealed decreased enrichment of the cell–cell adhesion pathway in adjacent normal samples. These components were primarily collagens and integrin interaction proteins, with phosphorylation on intercellular junction proteins such as Plakophilin-1 (S232) and Desmoplakin (S166). Overall, the corresponding findings in this analysis suggest that the gastric environment of BE patients is an important component of the molecular perturbations observed in our earlier analyses.

## 4. Discussion

Proteomic profiling is a powerful analytical technology for exploring global changes in the protein expression patterns and signaling pathways of transformed cells and pre-cancerous tissues [9]. In conjunction with modern systems biology approaches, we find that quantitative phosphoproteomics is especially well suited for comparative analyses of clinical specimens aimed at determining underlying biochemical alterations associated with pre-cancerous lesions, including metaplastic progression and early-stage pathogenesis, with the aim of identifying potentially actionable targets for improved clinical management. The establishment of a putative BE molecular signature based on our pilot study, supplemented by candidate signaling pathway-level analyses, highlights the utility of mass spectrometry-based profiling to measure molecular differences between non-dysplastic BE and normal tissue.

In this pilot study, we established the feasibility of a concerted proteomic and phospho-signaling pipeline to reveal molecular patterns that distinguish BE from normal epithelium, with the ultimate aim of informing on the fundamental mechanistic relationships underlying these distinct epithelial states. We evaluated the utility of a multi-step discovery and validation strategy for identifying a robust BE protein signature that is seemingly able to distinguish independent clinical specimens with good correspondence to disease status. Hence, despite the small initial pilot sample size and the well-documented heterogeneity of BE, these preliminary results suggest that the future scale-up of this approach could serve as a viable strategy to identify a robust set of markers that may hold promise for diagnosing and monitoring disease progression.

Our initial comparative pathway enrichment analyses revealed cellular processes that are seemingly significantly altered in BE. We mined these initial profiles for candidate functional insights, providing plausible initial descriptions of pathogenic mechanisms associated with non-dysplastic, metaplastic patient tissue. Notably, genomic maintenance was consistently altered in both the proteomic and transcriptomic pathway analyses (Appendix A). For example, we found cell cycle regulation to be consistently altered in BE, including decreased changes in the mitotic spindle checkpoint machinery. The Hyland transcriptomic study likewise showed significant alterations in this system (FDR = 0.021), as well as in related pathways such as the G2-to-M DNA damage checkpoint and mitotic sister chromatid segregation. There is strong evidence that DNA damage and poor regulation of the cell cycle are driving factors towards dysplasia and EAC [34,35], making DNA repair an important protective mechanism to prevent further genetic instability associated with tumorigenesis. Differences in the penetrance of these mechanisms could partly explain the variable progression seen in this disease.

Cell cycle progression and DNA repair depend on complex signaling mechanisms to ensure the maintenance of chromosome integrity during cell proliferation and are tightly regulated at the phosphorylation level. Thus, to parse phosphorylation-dependent activation of DNA repair and cell cycle regulation pathways, we performed motif enrichment analysis and found mechanisms enriched in our initial BE cohorts. BE-enriched cell cycle regulators included CSNK2A1 and UHMK1 (Appendix A), which are known to be upregulated in a number of cancers [36,37], as well as upstream mediators of p53-mediated apoptosis in response to the genomic damage response, such as PRKCD and CSNK2A1 [33] (Figure 4). Supporting this, BE lesions showed elevated phosphorylation of TP53-binding protein 1 (Appendix A), a key component of the ATM/RIF-1 double-strand break detection and repair system [32]. Gastric content exposure is known to cause oxidative base damage, and BE and EAC are reported to have high levels of oxidative DNA damage [38,39,40]. Tight regulation of single- and double-strand repair pathways may ensure genomic stability in the face of these chronic genotoxic insults and lesion progression.

We also noted epithelial-to-mesenchymal (EMT)-related alterations in BE (both in our proteome and the transcriptome studies), suggesting it as a potential early driver of disease development (Figure 3). EMT reflects a profound cytoskeleton reorganization resulting in the loss of cell adhesion, polarity, and increased extracellular matrix reorganization prior to metastasis in advanced malignant cancers [41]. The concomitant downregulation of epithelial development/differentiation pathways and the upregulation of extracellular matrix remodeling in our (phospho)proteome datasets, also seen in the Stairs transcriptome re-analysis, further supports a potential transition to a de-differentiated cellular state in pre-cancerous BE lesions (Appendix A). From our meta-analysis, it appears that BE lesions deviate from a well-defined differentiation state, predisposing the tissue to tumorigenesis, especially when coupled with other altered pathways, such as impaired cell cycle regulation and increased DNA repair.

Future elucidation of the phospho-signaling mechanisms and kinase–substrate associations illuminated by this pilot study has the potential to provide causal mechanistic insights, such as the role of activated DNA repair during the earliest pathological stages that precede tumorigenesis and genomic instability found in overt cancer. This feasibility study suggests that a larger-scale phosphoproteomic survey comparing BE versus normal epithelium has the potential to reveal causal mechanisms governing the cellular transformation and early-stage pathogenesis that precede dysplastic transformation and EAC. Our initial results suggest that defining the molecular state(s) that precedes dysplasia and overt cancer could serve as a resource for subsequent translational research.

## Figures and Tables

**Figure 1 genes-13-01215-f001:**
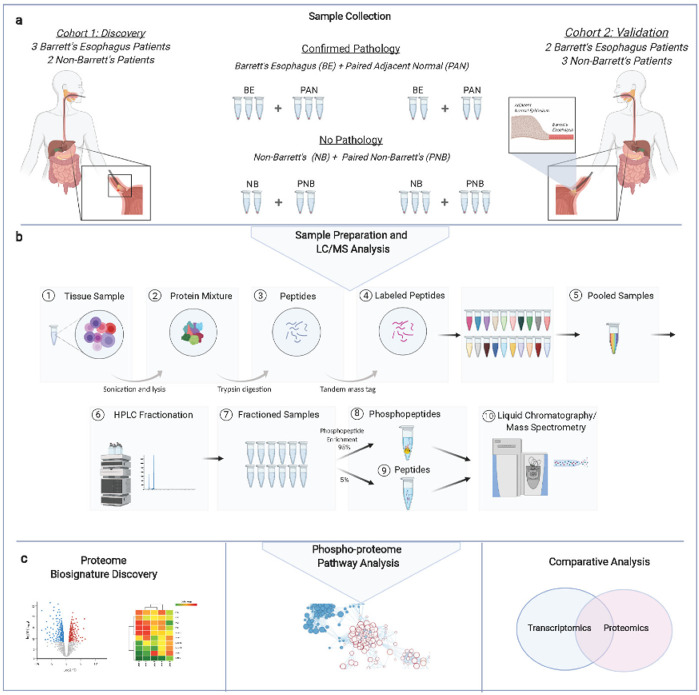
Schematic depictions of the pilot study workflow: (**a**) patient biopsy sample collection and categorization (BE = Barrett’s esophagus; PAN = paired adjacent normal; PNB = paired non-Barrett’s; NB = non-Barrett’s) assigned to two cohorts; (**b**) sample preparation pipeline for precision LC/MS analysis; (**c**) data analysis, starting from proteomic biosignature discovery (**left**), phosphoproteomics-based signaling and kinase–substrate associations (**middle**), and comparative analysis of differentially enriched pathways found in this study versus those obtained in independent transcriptomic studies of BE tissue (**right**).

**Figure 2 genes-13-01215-f002:**
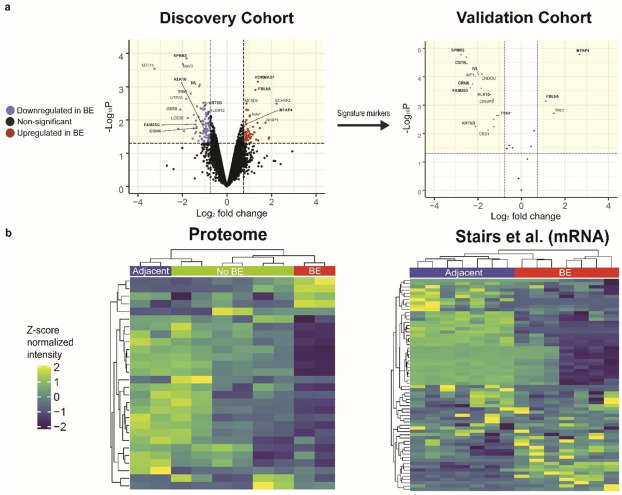
(**a**) Volcano plots (*p*-value vs. fold change) highlighting the preliminary Barrett’s esophagus signature proteins determined by quantitative proteomic analysis of paired BE/adjacent samples from the discovery cohort (**left**), which were then projected onto the validation cohort (**right**); highlighted sections represent significant threshold cutoffs (see main text); (**b**) heatmap displays showing hierarchically clustered BE, adjacent, and non-BE samples of the validation cohort based on the signature protein pattern (**left**) and classification of independent transcriptomes reported for BE and adjacent tissues by Stairs et al. based on cognate mRNAs matching our proteomics-based BE signature (**right**).

**Figure 3 genes-13-01215-f003:**
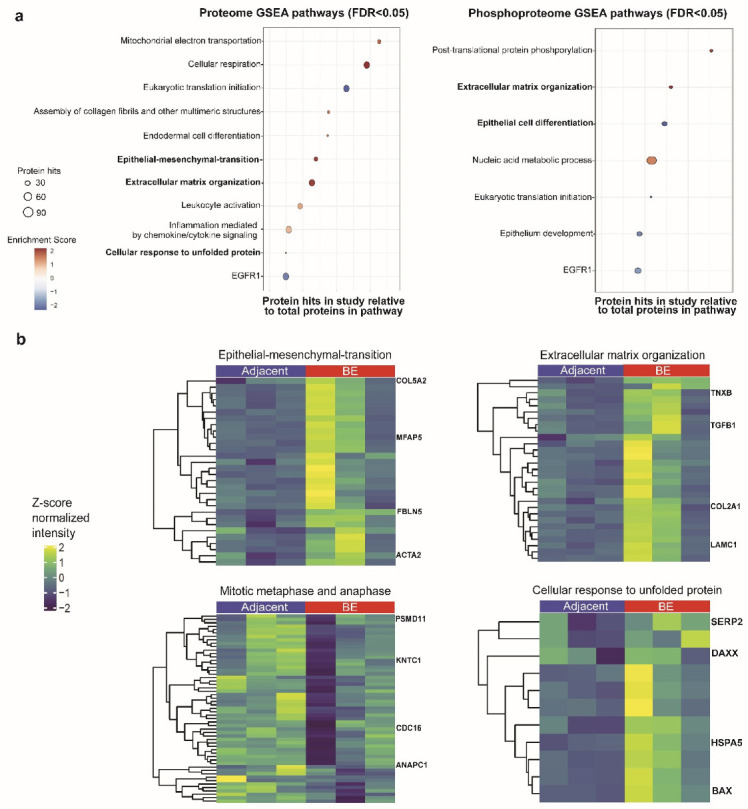
(**a**) Graphs of differentially regulated pathways (FDR < 0.05) in BE lesions relative to adjacent normal tissue, determined by GSEA. Selected sets of significantly enriched proteome (**left**) and phosphoproteome (**right**) pathways. Pathways are graphed as circles, in which the size denotes the number of feature hits detected in our study. Enrichment score denotes directionality, where a positive score indicates increased enrichment in BE and negative in adjacent normal. (**b**) Select subsets of GSEA proteome pathways highlighting important pathway proteins.

**Figure 4 genes-13-01215-f004:**
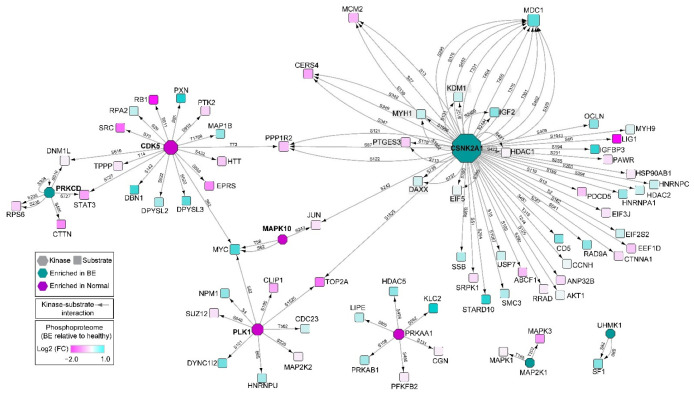
Protein kinase enrichment map showing differential kinase–substrate interactions inferred from motif (substrate) enrichment analysis of the tissue phosphoproteomic profiles. Implicated protein kinases (hexagons) and coherent changes in their putative substrate (circles) phosphorylation levels reflect the most significantly differential site-specific phosphorylation measurements captured by LC/MS in this pilot study. Phosphosites are listed with connecting edges, with the largest absolute log2-fold change in phosphorylation used for graphing substrates. Phosphosite analysis for kinases is displayed in Appendix Ab.

## Data Availability

ProteomeXchange (this study) Accession number: PXD023293; Reviewer account details: Username: reviewer_pxd023293@ebi.ac.uk; Password: gBfSdKtA; Hyland et al. Accession number: GSE39491; Nancarrow et al. Accession number: GSE28302; Stairs et al. Accession number: GSE13083.

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
