# Peer review of "Pilot Study Showing Feasibility of Phosphoproteomic Profiling of Pathway-Level Molecular Alterations in Barrett’s Esophagus"

_genes, 2022, doi:10.3390/genes13071215_

Round 1
Reviewer 1 Report
Congratulations to the authors on this exciting study. Moore et al. performed a pilot study evaluating the Phospho-Proteomic profiling in Barrett’s esophagus. Overall, the theme is relevant and may have importance for future research and clinical practice. I have some remarks.
The study objectives should be better depicted. The last paragraph of the introduction seems to be more appropriately localized in the discussion section.
In methods, the controls should be more detailed. How was the control group selected?
Authors could give more details on the studied group. What was the Prague classification of the included patients? Were the patients previously treated (ablation, fundoplication...)?
Author Response
Reviewer #1:
Congratulations to the authors on this exciting study. Moore et al. performed a pilot study evaluating the Phospho-Proteomic profiling in Barrett’s Esophagus. Overall, the theme is relevant and may have importance for future research and clinical practice.
REPLY: We are delighted that the reviewer found our study valuable towards both research and clinical practice.
MAJOR ISSUES
1) The study objectives should be better depicted. The last paragraph of the introduction seems to be more appropriately localized in the discussion section.
REPLY: We agree with the reviewer that this could be improved. We moved components of the last paragraph of the introduction to the discussion section, where they more appropriately summarize our findings, and replaced it with clearer explanations of the study objectives.
2) In methods, the controls should be more detailed. How was the control group selected?
REPLY: We addressed this concern by clarifying control group selection in the Materials and Methods section. Patients were enrolled into the control group based on prior diagnosis of Barrett’s Esophagus; patients with confirmed Barrett’s were placed in the Barrett’s Esophagus group while those without were placed in the non-Barrett’s control group. As a second internal validation and control within our study, we used adjacent esophageal epithelium that appeared normal to the endoscopist within the same patient. Both this specimen and the Barrett’s Esophagus were sent to the pathologist for reviewer for identification as either normal or Barrett’s Esophagus epithelium.
3) Authors could give more details on the studied group. What was the Prague classification of the included patients? Were the patients previously treated (ablation, fundoplication...)?
REPLY: For the patients who were identified and found to have BE epithelium, the classification was based on an academic pathologist understanding of the standard criteria that identifies Barrett’s Esophagus as distinct from normal epithelium. Previous treatment was not provided to protect patient privacy.
Reviewer 2 Report
Pilot Study Showing Feasibility of Phospho-Proteomic Profiling of Pathway-Level Molecular Alterations in Barrett’s Esophagus (genes-1746842)
The authors use proteomic analysis on human clinical biopsies to identify both proteins and phosphorylation events that are differentially expressed between normal esophageal mucosa and Barrett’s esophagus. A well-written paper, the authors provide proteomic datasets that should benefit the scientific community.
Major issues:
1. The impact of this manuscript is largely contained by the proteomic and phosphoproteomic datasets. However, a major issue in this paper, already acknowledged by the authors, is the small sample size. As a results FDRs are quite high and it is thus difficult to make any meaningful observations from the data. No new markers were identified, and Thus, I am puzzled why the authors chose to write up their study by dividing their data into a “Discovery Cohort” and a “Validation Cohort”. It is certainly a reasonable strategy for launching the study, but looking at the quality of the resulting data and the low number of signature markers, it seems much more useful to the scientific community to pool the two cohorts for all analyses.
2. The study design, comparing BE-affected tissue to normal margins, as well as the inclusion of tissue biopsies from non-affected healthy controls, are very good. However, I feel that the authors may not be taking full advantage of the valuable dataset they produced. Analyses could be made for example, between the healthy controls and the normal margins to identify potential early reactive changes to tissue injury from environmental factors (e.g. gastric reflux) which would discriminate between the normal margins from BE patients and healthy control subjects.
3. Heirarchical clustering (e.g. Figure 2b.) yielded unconvincing stratification between the three categories. Inclusion of something like PCA analysis would better answer the most basic question posed by this study: can proteomic and phosphoproteomic data distinguish between diagnostic categories?
Minor issues:
4. It seems impossible that a tissue sample of 250μg wet weight could yield 200μg of soluble protein (lines 97-102).
5. Use subscripts in chemical formulas. NH4HCO3 -> NH4CO3 (lines 103, 117).
6. Although the ratio of males to females is included, there is no mention of age. This is an important covariate and should be included (lines 158-9).
7. The statement “..we were able to capture multiple proteins associated with BE by prior comparative studies…” requires citations (line 178).
8. Enriched GO terms for EMT, does not “provide evidence of a non-dysplastic phenotype associated with tissue invasion.” The statement should be rewritten to something less definitive, using words such as “suggestive” etc. (lines 263-264).
9. GSEA does not identify direction of change, so cannot be said to trend “in the same direction”. Could be restated as affecting the same functional class of proteins (lines 305-306).
10. It is stated that phosphorylation-dependent signaling in BE by CSNK2A1 has links to important substrates such as MCM2 (lines 327-8), but yet, in Figure 4, MCM2 is displayed as being hypophosphorylated. How is that consistent with the suggested elevation in CSNK2A1 activity (implied) or protein levels?
11. Reference #12 (Korotkevich et al.) has an updated version in bioRXiv (line 522).
12. Reference #36 omitted the author list (lines 586-7).
13. Figure 4. A large number of protein kinases have phosphorylation-dependent kinase activity, therefore it would be useful to display phosphosite analysis of both kinases and substrates together.
14. In Supplementary Table 2, the values do not correlate with the figures. -Log10P values are completely different. Perhaps the -Log10P values were calculated for up-/down-regulation in adjacent tissue rather than BE?
Author Response
Reviewer #2:
The authors use proteomic analysis on human clinical biopsies to identify both proteins and phosphorylation events that are differentially expressed between normal esophageal mucosa and Barrett’s Esophagus. A well-written paper, the authors provide proteomic datasets that should benefit the scientific community.
REPLY: We are pleased that the reviewer finds that our work represents a significant for the community and well-written.
MAJOR ISSUES
1) The impact of this manuscript is largely contained by the proteomic and phosphoproteomic datasets. However, a major issue in this paper, already acknowledged by the authors, is the small sample size. As a results FDRs are quite high and it is thus difficult to make any meaningful observations from the data. No new markers were identified, and Thus, I am puzzled why the authors chose to write up their study by dividing their data into a “Discovery Cohort” and a “Validation Cohort”. It is certainly a reasonable strategy for launching the study, but looking at the quality of the resulting data and the low number of signature markers, it seems much more useful to the scientific community to pool the two cohorts for all analyses.
REPLY: We appreciate the suggestions and addressed these concerns two ways. First, we lowered the gene set enrichment analysis FDR threshold to FDR<0.05 for the discovery cohort analysis (proteome and phosphoproteome) to ensure robustness. Secondly, we added a new section and supplementary table which summarize the pooled datasets (gene set enrichment analysis of discovery and validation cohorts), titled “3.5. Pathway analysis of combined cohorts via proteomics and phosphoproteomics”. We noted important gene sets in this analysis, such as the unfolded protein response, as well as novel pathways related to cell division. As rightfully suggested by the reviewer, we were able to glean additional information from pooling these sets, and in result have a much stronger analysis.
2) The study design, comparing BE-affected tissue to normal margins, as well as the inclusion of tissue biopsies from non-affected healthy controls, are very good. However, I feel that the authors may not be taking full advantage of the valuable dataset they produced. Analyses could be made for example, between the healthy controls and the normal margins to identify potential early reactive changes to tissue injury from environmental factors (e.g. gastric reflux) which would discriminate between the normal margins from BE patients and healthy control subjects.
REPLY: We appreciate the reviewer’s recommendation and created a new section, titled “3.7. Non-BE and adjacent normal comparison reveal early reactive changes to environmental factors.”, which looks at these changes. We noted modest differences between the adjacent normal and non-Barrett’s samples. Importantly, our gene set enrichment analysis revealed similar, Barrett’s specific enrichment in the adjacent normal samples. We saw decreased enrichment of DNA repair and chromatin assembly pathways in the adjacent normal, as well as decreased cell-cell adhesion. Overall, these findings suggest that the gastric environment of Barrett’s Esophagus patients is an important driver of molecular changes seen in Barrett’s lesions.
3) Heirarchical clustering (e.g. Figure 2b.) yielded unconvincing stratification between the three categories. Inclusion of something like PCA analysis would better answer the most basic question posed by this study: can proteomic and phosphoproteomic data distinguish between diagnostic categories?
REPLY: We added a new supplementary figure containing the PCA plot for both the proteome and phosphoproteome of the discovery and validation cohorts together. This figure shows the separation between sample types, and we believe addresses this basic question. Nevertheless, our goal in this section was to generate a signature which distinguishes Barrett’s and adjacent normal samples, since these features can be used in future studies. Thus, we kept this portion of the study.
MINOR ISSUES
4) It seems impossible that a tissue sample of 250μg wet weight could yield 200μg of soluble protein (lines 97-102).
REPLY: The reviewer is correct. We incorrectly entered the mass (not measured) of the samples rather than the known volume of ~1.5 x 2mm in size. This has been amended.
5) Use subscripts in chemical formulas. NH4HCO3 -> NH4CO3 (lines 103, 117).
REPLY: The subscript format has been updated in all of these lines.
6) Although the ratio of males to females is included, there is no mention of age. This is an important covariate and should be included (lines 158-9).
REPLY: Thank you for this suggestion. While we agree that this is an important covariate, this information was not provided to protect patient privacy.
7) The statement “..we were able to capture multiple proteins associated with BE by prior comparative studies…” requires citations (line 178).
REPLY: We agree with the reviewer’s recommendation and have added the appropriate citations.
8) Enriched GO terms for EMT, does not “provide evidence of a non-dysplastic phenotype associated with tissue invasion.” The statement should be rewritten to something less definitive, using words such as “suggestive” etc. (lines 263-264).
REPLY: We agree and opted to tone down the language in our original claims. Enriched GO terms provide evidence of particular cell processes, and no definitive proof. We changed our language to “… suggesting an association of this non-dysplastic phenotype with tissue invasion” to make this clear.
9) GSEA does not identify direction of change, so cannot be said to trend “in the same direction”. Could be restated as affecting the same functional class of proteins (lines 305-306).
REPLY: This is true. To clarify that we were saying that the same pathways and directionality of normalized enrichment score is seen between studies, we change the sentence to: “Notably, the differential expression of these pathways generally shared similar normalized enrichment scores as seen with the proteomics GSEA”. We hope this is more accurate.
10) It is stated that phosphorylation-dependent signaling in BE by CSNK2A1 has links to important substrates such as MCM2 (lines 327-8), but yet, in Figure 4, MCM2 is displayed as being hypophosphorylated. How is that consistent with the suggested elevation in CSNK2A1 activity (implied) or protein levels?
REPLY: In Figure 4a, we graph all phospho-sites on the substrates (e.g. MCM2 has S138, S27, and S13), and only the phosphosite with the highest absolute is used to color substrates. In the case of MCM2, S139 (log2FC of -0.51) is graphed since it has the highest value and thus showing the substrate as hypo-phosphorylated, while S27 shows increased phosphorylation in Barrett’s Esophagus samples. We chose to graph the sites this way in order to maintain simplicity and clarity to the figure. We clarified this in the caption by stating “Phosphosites are listed with connecting edges, with the largest absolute log2-fold change in phosphorylation used for graphing substrates”.
11) Reference #12 (Korotkevich et al.) has an updated version in bioRXiv (line 522).
REPLY: We thank the reviewer for the suggestion. This reference has been updated.
12) Reference #36 omitted the author list (lines 586-7).
REPLY: This reference has been updated.
13) Figure 4. A large number of protein kinases have phosphorylation-dependent kinase activity, therefore it would be useful to display phosphosite analysis of both kinases and substrates together.
REPLY: We agree, this is important information to display. We originally chose not to show kinase level phosphosite analysis to maintain simplicity of the figure, but to address this we amended Supplementary Table 6b. We highlighted and annotated specific kinase phosphosites, and added to the caption of Figure 4 to direct readers to this information.
14) In Supplementary Table 2, the values do not correlate with the figures. -Log10P values are completely different. Perhaps the -Log10P values were calculated for up-/down-regulation in adjacent tissue rather than BE?
REPLY: Thank you for pointing out this discrepancy. Before we run differential analysis, we Log2 transform and z-score normalize the raw intensity values. When we analyze the Barrett’s Esophagus, adjacent normal, and non-Barrett’s Esophagus samples together we get different z-score normalized values from when we just analyze Barrett’s Esophagus and adjacent normal (see expression values in Supplementary Table 2a vs Table 4a). The p-value is changed according to these different relative expression values. We calculated the -Log10(p-value) in Figure 2a from supplementary table 4a, which only includes the adjacent and Barret’s samples.